# Development of a Clean Label Mayonnaise Using Fruit Flour

**DOI:** 10.3390/foods12112111

**Published:** 2023-05-24

**Authors:** Maria Rocha Vieira, Sara Simões, Cecilio Carrera-Sánchez, Anabela Raymundo

**Affiliations:** 1LEAF—Linking Landscape, Environment, Agriculture and Food Research Center, Associated Laboratory TERRA, Instituto Superior de Agronomia, Universidade de Lisboa, Tapada da Ajuda, 1349-017 Lisbon, Portugal; mariarvieira98@gmail.com (M.R.V.); sfrsimoes@isa.ulisboa.pt (S.S.); 2Departamento de Ingeniería Química, Escuela Politécnica Superior, Universidad de Sevilla, Calle Virgen de África, 7, 41011 Sevilla, Spain; cecilio@us.es

**Keywords:** mayonnaise, clean label, food by-products, nectarine flour, antioxidant capacity

## Abstract

Over the past few years, clean label food has been growing, meaning that consumers are searching for shorter and simpler ingredient lists composed of familiar and natural ingredients. The objective of the present work was to develop a vegan clean label mayonnaise, replacing the additives with fruit flour obtained from fruit reduced commercial value. The mayonnaises were prepared by replacing the egg yolk with 1.5% (*w*/*w*) lupin and faba proteins, while fruit flour (apple, nectarine, pear, and peach flour) was incorporated to substitute sugar, preservatives, and colorants. Texture profile analysis and rheology—small amplitude oscillatory measurements were performed to evaluate the impact of the fruit flour on mechanical properties. The mayonnaise antioxidant activity was also analyzed in terms of color, pH, microbiology, and stability measurements. The results showed that mayonnaises produced with fruit flour had better structure parameters in terms of viscosity, and texture, but also improved pH and antioxidant activity (*p* < 0.05) compared to the standard mayonnaise (mayonnaise without fruit flour). The incorporation of this ingredient into mayonnaise increases the antioxidant potential, though it is in lower concentrations compared to the fruit flours that compose them. Nectarine mayonnaise showed the most promising results in terms of texture and antioxidant capacity (11.30 mg equivalent of gallic acid/100 g).

## 1. Introduction

Currently, consumers are looking for more natural, healthy, and sustainable processed foods. Due to this demand, producers focus on reformulating traditional products with ingredients that meet consumers’ demands. Two trends appeared to support consumers’ new desires, which are plant-based and clean label products.

According to Park & Kim (2021) [1], the market for clean label ingredient foods is expected to develop significantly, reaching nearly 47.50 billion dollars by 2023, with a compound annual growth rate of 6.8% throughout the forecast period. However, there is no legal definition regarding this subject, leading to the creation of different definitions [2]. For example, the Institute of Food Technologists suggests that a clean label product is made with as few ingredients as possible and contains ingredients that customers understand and consider to be healthy [3]. As a result, “free from additives,” “minimal processing,” and “short ingredient listing” are essential requirements for food to be given a clean label [4].

Mayonnaise is an oil-in-water (*o*/*w*) emulsion that traditionally contains 65–80% fat, providing the typical texture, appearance, and shelf-life of this product. Due to its mouthfeel and particular flavor, it is consumed as a traditional seasoning [5]. It is probably one of the most widely used sauces worldwide, as more than 6.5 million people reported consuming it more than once a week in the UK in 2018, as well as 4.5 million in France, and 3.4 million in Spain [6]. Mayonnaise is usually composed of an emulsifier (egg lecithin), vegetable oil, acid components (vinegar, citric acid, and maleic acid), texture enhancers, flavoring ingredients, and stabilizers [5].

Although egg yolk is well-known for its emulsifying properties, there are some inconveniences with using this animal source. Hen’s egg is a common food allergen but is also a source of possible contamination with *Salmonella* sp. [7]. Moreover, it has unfavorable high cholesterol contents, which can lead to health problems [8,9]. Alongside this, there has been an exponential increase in people turning vegan, i.e., completely eliminating animal products from their diet and consumption habits, with the number of individuals identifying themselves as vegan rising 600% in the US between 2014 and 2017 and quadrupling in the UK between 2014 and 2018 [6]. For that reason, the industry has developed alternatives using vegetable protein. Egg is commonly substituted into foods such as mayonnaise, meringue, and pasta, using ingredients derived from legumes [6]. There are numerous vegan egg substitutes on the market, which are made from diverse sources such as soy or pea protein isolates, potato starch, or chickpea flour [6]. Replacing egg yolk with vegetable protein provides several advantages, such as increasing microbiological stability, lowering manufacturing costs since there is no need for pasteurizing eggs, and reducing cholesterol content [10].

According to FAO, per year, around 1.3 billion tons of food are wasted globally, representing one-third of the total food sector production [11]. The food category that contributes most to this cause is fruits and vegetables, representing about 0.5 billion tons of waste due to their perishable nature but also due to not meeting aesthetic criteria such as size, shape, or color [12,13]. Hence, food manufacturers need to find a way of incorporating fruit by-products again into the food chain, contributing to a reduction in generated waste and to a more circular economy [9,12].

Fruit flour is regarded as a clean label ingredient, which can be applied as a food additive in food products [12]. These flours can be produced from fruits without commercial value and by-products of the manufacture of fruit products. Thus, processing them into flour allows its valorization and incorporation into the chain in the logic of circular economy [14]. This ingredient provides health benefits due to its high fiber and antioxidant content. Through its use, it is possible to incorporate bioactive compounds, such as carotenoids and phenolic acids in mayonnaises, increasing the stability and shelf-life of the final products, while enhancing the product’s color and increasing potential health benefits [13]. A potential texture-enhancing capacity is expected due to its dietary fiber content, which impacts the water-holding and retention capacity, oil-holding capacity, foam capacity, and stability [13]. Adding fruit flour to liquid or semi-solid food products will improve their viscosity, even when added in small amounts [13]. Along with these advantages, fruit flours have important technological functionalities such as using them as thickeners, gelling agents, fillers, and water retainer agents, but also in the production of edible films [13].

The present work was integrated into the cLabel+ project and developed in partnership with the company Casa Mendes Gonçalves. The main objective was the development of a clean label vegan mayonnaise, characterized by the absence of additives, such as preservatives, sugars, and colorants, using a clean label ingredient to substitute them. To achieve this, vegetable proteins, proposed by Mendes Gonçalves S.A, were used to replace the egg yolk in mayonnaise (work previously developed [15]) and national fruit flours, obtained from the circular economy, were incorporated as a clean label ingredient. A standard commercial egg yolk mayonnaise produced by the company and a previously developed vegan mayonnaise [15] were considered as targets.

## 2. Materials and Methods

### 2.1. Materials

All ingredients used are commercially available and were provided by the suppliers of the company, including, refined sunflower oil, potassium sorbate >98% (*w*/*w*) purity, fine granulated sugar, refined salt, lemon juice (concentrated by a factor of 7), rosemary extract, 8–10% (*v*/*v*) acetic acid alcohol vinegar, β-carotene colorant, and vegetable proteins (lupin protein isolate with 87–95% protein content, and faba bean protein concentrate with 60% protein content). To produce four different fruit mayonnaises, apple, pear, nectarine, and peach flours were produced by AgroGrIN Tech (presented in Table 1). Concerning the production of Mendes Gonçalves commercial mayonnaise (MG), pasteurized egg yolk and modified starch were used.

All fruit mayonnaises were later compared to standard mayonnaise (mayonnaise without fruit flour) and with MG produced on a laboratory scale, to have the same emulsifying scale. The composition of the three different types of mayonnaises is shown in Table 2.

### 2.2. Mayonnaise Preparation

For the development of the present work, standard mayonnaise was produced based on Cabrita et al.’s, (2022) [15] formulation, in which the vegetable protein system is composed of 50% lupin and 50% faba protein. Fruit flour was added to directly replace sugar, potassium sorbate, and β-carotene. Before producing both mayonnaises, proteins were hydrated in demineralized water for 30 min under magnetic stirring at high speed. For standard mayonnaise, this step consisted of mixing the water with colorant, and, gently, small portions of protein were added at a time, whereas in the fruit mayonnaises, the fruit flour was mixed with the proteins. A lab-scale mixer, Ultra-Turrax T-25 (IKA-Werke GmbH & Co. KG, Staufen im Breisgau, Germany) homogenizer, was used to mix all the ingredients together in a three-step process in order to keep a closely packed emulsion [16]: (i) powders were added all at once under the agitation of 6000 rpm until achieving a homogenous mixture; (ii) the speed was increased until 9000 rpm and slowly the sunflower oil (65% *w*/*w*) and rosemary extract (also a clean label ingredient) were added together to the mixture with stirring for 6 min until homogeneous; (iii) the acids (vinegar and lemon juice concentrate) were added simultaneously at 11,000 rpm for 2 min. Mayonnaise samples were stored in cylindrical glass jars (100 mL with 66 mm diameter, 56 mm height) and kept in the fridge at 4 °C for 24 h to achieve equilibrium.

### 2.3. pH Determination

The sample pH was measured at room temperature using a probe for solid samples pH-Meter BASIC 20 (Crison Instruments, S.A., Alella, Spain) with an electrode previously calibrated that was inserted into the center of the sample, where three replicates of each sample were taken.

### 2.4. Color Measurements

Instrumental color evaluation was performed using a Chroma Meter CR-400 (Konica Minolta Business Technologies, Inc., Tokyo, Japan), based on the CIELAB color coordinate system (*L**, *a**, *b**). For each sample, the color was evaluated at six different points, following equipment calibration against the white plate. The total color change (Δ*E*) was calculated for each sample using the following formula [17]:(1)$$∆E^{*}=\sqrt{{\left(L_{i}^{*}-L_{0}^{*}\right)}^{2}+{\left(a_{i}^{*}-a_{0}^{*}\right)}^{2}+{\left(b_{i}^{*}-b_{0}^{*}\right)}^{2}}$$
where the abbreviation of color parameters represents 0-standard mayonnaise and i-remaining mayonnaises. When the Δ*E** value is superior to 5 units, it indicates that the human eye is capable of perceiving differences between the colors of 2 samples [18].

### 2.5. Rheological Measurements

The rheological characterization of mayonnaise was performed in a controlled-stress rheometer, Haake MARS III (Thermo Fisher Scientific, Waltham, MA, USA), equipped with a UTC Peltier. A CC35_2Ti cone-and-plate sensor system (35 mm diameter, 2°) was used to perform small amplitude oscillatory shear measurements (SAOS) in a frequency range of 0.01–100 Hz, within the linear viscoelastic region that was previously assessed. Thus, it was possible to obtain the mechanical spectrum, such as the storage modulus (G′) and loss modulus (G″) as a function of frequency.

G′ at 1 Hz and the loss tangent were also collected. Plateau Modulus (G^0^_N_) is achieved by tan δ (tan δ = G″/G′), giving information about the contribution of the protein films entanglements that coat the droplets to the structure of the system. This corresponds to the value of G′ obtained when tan δ reaches the minimum in the frequency range studied [19].

In order to obtain flow curves, a shear rate of 10^−8^ to 500 s^−1^ was applied, using a serrated parallel plate system PP20 (20 mm of diameter, rough surface). These types of plates were used so that the sample did not slip [20]. Since frequency and viscosity tests are extended tests, paraffin oil was placed around the plate/cone and plate to avoid evaporation of the sample during measurements. Each rheological test was conducted at a controlled temperature of 20 ± 0.5 °C in a temperature-controlled room with previous stabilization of the samples for half an hour. The Origin 2019 software (OriginLab, Northampton, MA, USA) was later used to align the viscosity versus shear rate curve to the Williamson model, as Álvarez-Castillo et al. (2021) [21] used.
(2)$$\eta =\frac{\eta_{0}}{1+{\left(k\times \dot{\gamma }\right)}^{n}}\;$$
where *η*_0_ is the zero shear rate limiting viscosity at low shear rates (Pa.s); *k* is the consistency coefficient (Pa.s), and *n* is a dimensionless shear-thinning index, which is the slope of the power-law shear-thinning region, with *n* < 1 for shear-thinning fluids.

### 2.6. Textural Profile Analysis

The Texture Profile Analysis (TPA) was carried out on a TAX plus texturometer (Stable Micro Systems, UK) with a load cell of 5 kg and a 19 mm diameter Perpex cylindrical probe. This test was performed at 20 °C in a temperature-controlled room with previous stabilization of the samples for one hour before each replica. The test proceeded with the penetration through the sample for 15 mm at a speed of 1 mm/s, to subsequently return to its original position, where the time between two bites was 5 s. Mayonnaise samples were evaluated on full cylindrical glass jars (100 mL with 66 mm diameter, 56 mm height). The test was run in five replicates, obtaining, in the end, a force versus time texturograms, where firmness and adhesiveness were collected.

### 2.7. Emulsion Structure and Stability

#### 2.7.1. Droplet Size Distribution (DSD)

The mean droplet diameter of the emulsions was obtained using a laser diffraction instrument—Mastersizer 2000; Malvern Instruments, UK. Samples were previously dispersed in distilled water and stirred before determination to ensure homogeneity. Pre-dispersed samples were added until 6 to 20% of the obscuration range was achieved. The refractive index used was 1.46, corresponding to sunflower oil [22]. In order to determine the existence of flocculation on the studied samples, DSD was also obtained in the presence of Sodium Dodecyl Sulphate (SDS) (ITW reagents), a surfactant known to disrupt floccules formed in emulsions [23]. The presence of flocculation in the systems studied is determined by the different profiles obtained either in the presence or absence of SDS [24].

Sauter mean diameter (*d*_3,2_) expresses the mean diameter for most of the droplets and can be obtained through the following expression:(3)$$d_{3,2}=\frac{\sum n_{i}d_{i}^{3}}{\sum n_{i}d_{i}^{2}}\;$$
where *n_i_* is the number of droplets with a diameter *d_i_*.

The volumetric diameter (*d*_4,3_) is associated with modifications in particle size as the result of several destabilizing mechanisms, being more susceptible to fat droplet aggregation [25], and may be obtained through the equation:(4)$$d_{4,3}=\frac{\sum n_{i}d_{i}^{4}}{\sum n_{i}d_{i}^{3}}\;$$

All the DSD measurements were carried out at room temperature, at least in triplicate, on fresh samples (day 1) and up to 30 days after emulsion preparation. From these measurements, the respective values of the diameters *d*_4,3_ and *d*_3,2_ were obtained with and without SDS.

#### 2.7.2. Backscattering

Emulsion stability was assessed using a vertical scan analyzer Turbiscan MA 2000 (Formulaction, France), through backscattering measurements of a pulsed light source (λ = 850 nm) as a function of the height of the cylindrical glass tube containing the sample [25]. The mayonnaises were disposed in 5 mL glass jars (radius 27.5 mm, height 70 mm) up to 40 mm height, stored at 5 °C, and measured at room temperature for 30 days after emulsion formation. Then, the profiles of backscattering of light from emulsions (ΔBS/100%) versus the height were plotted as a function of storage time [22].

#### 2.7.3. Visible Light Microscopy

The analog microstructure of mayonnaise was analyzed with a Leica-SP2 (Germany) confocal optical microscope equipped with a 4 MP digital camera. A total of 10 μL of the prepared samples was placed directly on top of a transparent slide (76 × 26 mm) gently coated by coverslips (18 × 18 mm). To avoid drying, samples were observed directly, obtaining images using an objective of 100 magnifications (100×).

### 2.8. Antioxidant Potential

#### 2.8.1. DPPH Method

The DPPH technique was performed following the method described by Bunzel & Schendel (2017) [26], where a solvent extraction was conducted primarily. In this case, 5 g of sample were added to 15 mL of methanol (Labchem, Zelienople, PA, USA) and shacked for 1 h with Reax 2 *w*/*o* adapter (Heidolph, Schwabach, Germany) with a rotation speed range of 3 rpm. After that, for 10 min at 4 °C, and for a speed of 4000 rpm a centrifuge was performed on a HERMLE Z 383 K (Wehingen, Germany). A total of 100 μL of the sample’s supernatant was added to 2.4 mL of a 103.5 μM of methanol solution of DPPH (Alfa Aesar, Ward Hill, MA, USA) and mixed with a Vortex. This procedure was made in triplicate and the blank samples were only prepared with methanol and sample. After 1 h in the dark, at room temperature, the absorbance of the samples was read in a Cary Series UV-Vis spectrophotometer (Agilent Technologies, Santa Clara, CA, USA) at 517 nm. A standard curve using standard solutions of gallic acid (Sigma Aldrich, St. Louis, MO, USA) with methanol was performed in the range of concentrations between 0 and 0.1 mg/mL.

#### 2.8.2. FRAP Method

A fat extraction, identical to the one performed in the DPPH method, was realized before carrying out the FRAP assay. The method described by Benzie & Strain (1996) [27] was followed with slight modifications. The FRAP reagent was prepared by mixing 100 mL of 300 mM acetate buffer (pH = 3.6), 10 mL of 20 mM FeCl_3_·6H_2_O, and 10 mL of 10 mM 2,4,6-tripyridyl-s-triazine (TPTZ) solution in a 10:1:1 ratio. The TPTZ solution was obtained by mixing 10 mM TPTZ (Alfa Aesar) in 40 mM HCl (HoneyWell, Charlotte, NC, USA). In test tubes, 90 μL of sample supernatant with 270 μL of deionized water and 2.7 mL of FRAP solution (except on the control samples) were added. Each sample was run in triplicate. For 30 min, the test tubes were left in a water bath at 37 °C to react, and then the absorbance was read in a Cary Series UV-Vis spectrophotometer (Agilent Technologies, Santa Clara, CA, USA), at 595 nm. To determine the FRAP value of each sample, a subtraction between FRAP reagent with the sample and an initial blank reading with just deionized water and sample was performed. The results were compared with a gallic acid calibration curve, with a concentration between 0 and 0.25 mg/mL.

### 2.9. Total Phenolic Compounds

Before starting the Total Phenolic Compounds (TPC) determination, a solvent extraction identical to the one performed in the DPPH method was carried out. The method used was performed following the one stated by Bunzel & Schendel (2017) [26], and conveniently modified. All mayonnaise and fruit flour extracts were analyzed photometrically by virtue of the Folin–Ciocalteau reagent. In test tubes, 100 μL of sample supernatant along with 450 μL of Mili-Q water and 50 μL of Folin’s reagent (PanReac AppliChem, Darmstadt, Germany) were mixed, followed by a mixture in the vortex. After 3 min of reaction, 400 μL of Na_2_CO_3_ (7.5% *m*/*v* HoneyWell) solution was added. The mixture was kept out of sunlight for 2 h at room temperature. TPC was read in a Cary Series UV-Vis spectrophotometer (Agilent Technologies, Santa Clara, CA, USA) at 760 nm. The analyses were performed in triplicate and the results were expressed as gallic acid equivalent (mg equivalent of gallic acid/100 g of product). The quantification was carried out by performing a standard curve using standard solutions of gallic acid in the range of concentrations between 0 and 0.5 mg/mL.

### 2.10. Microbiological Analysis

To determine the different types of microorganisms that might be present in mayonnaise samples, different microbiological analyses were performed and are synthesized in Table 3.

### 2.11. Statistical Data

The results obtained were submitted to statistical analysis for verifying the occurrence of significant differences between samples. To conduct the analysis of variance (ANOVA), a statistical analysis program, GraphPad Prism Software (version 5.0) was used. In this program, the Tukey test was used to compare more than 2 samples, with a 95% degree of confidence (*p* = 0.05). Differences were considered significant when *p* values were inferior to 0.05.

## 3. Results and Discussion

### 3.1. Addition of Fruit Flour in the Mayonnaise

The work carried out promoted the introduction of fruit flour (apple, nectarine, pear, and peach) to replace sugar, potassium sorbate, and β-carotene colorant. Hence, 0.63% of fruit flour was introduced, which represents the amount that once existed of the previously mentioned ingredients.

These mayonnaises (Figure 1) were later compared with standard mayonnaise and with MG made on a laboratory scale.

#### 3.1.1. pH and Colour Evaluation

The results obtained from ANOVA show that the addition of fruit flour had a significant effect on the pH value of mayonnaise (*p* < 0.05), reducing its value (Table 4). The pH of the mayonnaise has an impact on the structure and stability of the emulsion [28]. Resulting from the addition of fruit flour, all mayonnaises, especially APPM and PCHM mayonnaises, were similar to MG, which was not achieved without this incorporation. By decreasing the pH value, fruit flour permits an extension of the product’s shelf-life. This decrement can be explained by the high amount of pectin content that these flours have. Pectin is a carboxylic group and is acidic. Moreover, this component is also a D-galacturonic polymer linked by -1,4 glycosidic bonds which play a role in lowering pH [29]. Mirazimi et al. (2022) [30] have also concluded that increasing the percentage of pear flour in yogurts decreases the pH value of the product.

The CIELAB colour parameters, as well as Δ*E** values, are presented in Table 4. As for the *L** parameter, lower values indicate that samples have lower luminosity. Thus, it was found that the addition of fruit flour darkens the samples, except for NCTM, which does not have significant differences from the standard one (*p* > 0.05). The *a** parameter reveals a tendency to red for an *a*+ or green for an *a*−. The fruit flours kept the mayonnaise in a greenish tone, but decreased its values, bringing them closer to the reddish tone. This may happen due to the high number of carotenoids that these fruit flours impart [13]. *b** parameter reflects the tones of yellow (*b*+) and blue (*b*−). The addition of fruit flour reduced the values of this parameter; nevertheless, they are still in the yellow tones. These values might have been reduced since the majority of the yellowness of mayonnaise comes from egg yolk and oil [31]. Thus, fruit mayonnaises, by not having egg yolk in their composition, decreased the *b** parameter. Worrasinchai et al. (2006) [32] reported that there was a decrease in the *L** value of mayonnaise samples when adding β-glucan as a fat replacer, except for the mayonnaise with a reduction in fat content of 50%. Additionally, they found out that increasing the β-glucan ratio in the mayonnaise formula increased the *a** value while decreasing the *b** value. Maneerat et al. (2017) [33], on the other hand, found that after incorporating pectin extracted from banana peels into the mayonnaise formula as a fat replacer, the colour of salad cream became redder, less yellow, and darker, like in the present study.

Comparing the total colour difference (Δ*E**) of standard mayonnaise with the others, only MG is distinguished by the naked eye. The one with the least differences from standard mayonnaise is APPM. On the other hand, all mayonnaises are distinguished by the naked eye when compared to MG, since Δ*E** values are greater than 5 [18].

#### 3.1.2. Rheological Measurements

Figure 2 shows the evolution of storage (G′) and loss modulus (G″) with frequency obtained for each mayonnaise at 20 °C.

The shape of the curves indicates a predominantly elastic behavior since all emulsions show an elastic modulus (G′) higher than viscous modulus (G″) for the whole range of frequency. Considering that the two modulus of all samples do not stay apart for more than a decade, they present a weak gel-like behavior. Moreover, all the mechanical spectra present a similar slope and pattern, meaning that all the emulsions studied have similar viscoelastic properties in the experimental range of frequencies. Similar behavior was found for vegetable protein-stabilized emulsions by other authors [19,34,35]. From Figure 2 it is also possible to observe that PCHM and PRM show the highest viscoelastic modulus values, whereas standard mayonnaise shows the lowest.

To complement the analyses of the preceding figure, Table 5 shows the rheological parameters of these mayonnaises. The values of the plateau modulus and G′ at 1 Hz of all samples are congruent with the previous results, that is, samples with a higher degree of structuring (viscous and elastic modulus further) have higher values in these parameters. When observing the values of G′ at 1 Hz, there were no significant differences (*p* > 0.05) between PCHM and PRM, and they were the most structured. The same happens with MG and standard mayonnaises (*p* > 0.05) but with the lowest values. It was found that the sample with the greatest plateau modulus, and therefore with the highest degree of structuring, were PRM and PCHM. Thereby, the flours, in general, led to an increase in the plateau module, and since it is related to the stability of emulsions, this ingredient made the mayonnaises more stable. Bozdogan et al. (2022) [36] reported that by adding Pear Pomace Powder (PPP) to quinoa-based gluten-free cakes, the viscoelastic properties of batters increased with the amount of PPP added to the sample, having greater results in comparison with the control sample, which is consistent with the present work. Likewise, Kırbaş et al. (2019) [37] found that apple, carrot, and orange pomace powders have an increasing influence on dough viscoelastic characteristics compared to the control sample.

Figure 3 illustrates the flow behavior of mayonnaises, where it is possible to notice that all samples present the behavior of flocculated emulsions since at a low shear rate the samples present a Newtonian behavior characterized by a constant viscosity (*η*_0_). Subsequently, with the decrement in velocity, the shear rate increases, with it being possible to observe the shear-thinning region [38].

All samples show limiting viscosity values in the same order of magnitude (Figure 3). The limiting viscosity (*η*_0_) values are very close for the different flours, not showing a direct relationship with the linear viscoelastic parameters: the samples with higher plateau modulus values did not present a higher degree of sample structuring. Hence, the sample with higher apparent viscosity values is MG, which presents significant differences compared to the remaining samples (*p* < 0.05). Standard mayonnaise remains the one with the lowest values.

The increase in viscosity, resulting from the addition of fruit flour, may be related to the high content of fiber that flours have, such as pectin which acts as a thickening agent. Moreover, Mistrianu et al. (2022) [39] found out that by adding beetroot by-product powder to mayonnaise samples, the viscosity of the mayonnaise was greatly enhanced, contributing to the development of a creamy consistency. Similar results were observed by other authors [40,41], in which all samples of mayonnaise with processed beetroot and watermelon rind flour presented higher viscosity than the control sample.

#### 3.1.3. Texture

In Figure 4, the average results of firmness and adhesiveness obtained from TPA tests are represented. These texture parameters were the ones that best discriminated the samples under study.

Regarding the results of firmness, the sample with the greatest value is MG, with a significant reduction in this parameter (*p* < 0.05) in the remaining samples. When comparing only the fruit mayonnaises with the standard, there was a slight increase in firmness, where NCTM stands out with the greatest result among fruit mayonnaises. Mistrianu et al. (2022) [39] also obtained an increase in firmness after the introduction of beetroot peels powder in mayonnaise, a phenomenon that is explained by the introduction of stabilizing compounds, such as pectin, in the formulation.

As for the adhesiveness results, it is observed that the MG remained the sample with the highest values and, again, NCTM showed up a higher adhesiveness value compared to the standard mayonnaise. Lucera et al. (2018) [42] also noted an enhancement of the adhesiveness value of the final product due to the incorporation of vegetable flours on spreadable cheese.

#### 3.1.4. Antioxidant Capacity and Total Phenolic Compounds

From Table 6, it is possible to observe the antioxidant potential (DPPH and FRAP) and total phenolic compounds (TPC) of fruit flours and the ingredients that they replaced (sugar, potassium sorbate, and colorant) represented by mg equivalent of gallic acid/100 g.

Comparing the fruit flours with the ingredients present in the standard mayonnaise (STF), it is possible to understand that the first has a significantly higher (*p* < 0.05) amount of total phenolic compounds and higher antioxidant capacity. Pear flour (PRF) is the flour with the highest levels of antioxidant potential; nonetheless, the content of phenolic compounds does not show significant differences (*p* > 0.05) between this flour and the remaining fruit flours. The lower antioxidant potential and phenolic compounds content is exhibited by STF.

By adding fruit flour to the mayonnaise formulation, the antioxidant potential increased when compared to the standard mayonnaise (Table 7). The enhancement in the antioxidant potential of the standard mayonnaise, compared to the ingredients that are used on this mayonnaise, can be attributed to several factors, such as: (i) the presence of antioxidants in the oil and other ingredients, such as rosemary extract [43]; (ii) their stability; or (iii) the method utilized to determine TPC, i.e., the Folin–Ciocalteau method, since this approach is not selective, leads to an identification of various reducing chemicals that react with the Folin–Ciocalteau reagent [42].

There is also a matrix effect since significant differences detected in flours (Table 6) are no longer noticeable when introduced in mayonnaise (Table 7). Still, the antioxidant effect in mayonnaises with fruit flour is higher than the standard mayonnaise (*p* < 0.05). These findings were also reported by Lucera et al. (2018) [42] when flours from vegetable and fruit by-products were used to fortify spreadable cheese. Red grape pomace flour (107.40 ± 2.08 mg GAEs/g dw) and artichoke by-product flour (21.15 ± 0.24 mg GAEs/g dw), after being incorporated into the cheese lost more than half their antioxidant capacity, 1.20 ± 0.22 mg GAEs/g dw and 2.34 ± 0.15 mg GAEs/g dw, respectively. Nonetheless, the TPC and flavonoid content increased significantly, compared to the control.

When analyzing FRAP, there are no significant changes among mayonnaises (*p* > 0.05), whereas DPPH analysis shows that NCTM and PCHM mayonnaises are similar to the standard mayonnaise. Not only in FRAP but also in DPPH results, PRM is the one that presents higher antioxidant potential, being in accordance with the results shown in Table 7. Finally, when evaluating the samples through the Total Phenolic Compounds they do not present significant differences (*p* > 0.05).

Ishartati et al. (2019) and Espinosa-Solis et al. (2019) [37,38] found that by adding apple flour to cookies and pasta, respectively, the antioxidant activity of the final products increased. This is explained by the high content of bioactive compounds in particular anthocyanins, phenolic acids, dihydrochalcones, and flavanols that apples contain [44]. Evanuarini & Susilo (2021) [29] produced a mayonnaise with the addition of apple peel flour, revealing that the addition of this ingredient gave a highly significant effect on the antioxidant activity of mayonnaise (8.70–19.38%), with the mayonnaise with 3% apple peel flour being the one with the highest antioxidant activity.

#### 3.1.5. Droplet Size Distribution (DSD) and stability

Among the four fruit flours previously analyzed, the one that was more akin to the standard and MG was selected to continue the studies. In this way, nectarine flour was chosen, and stability measurements were performed.

In order to have a better understanding of the stability of the emulsions, analyses of the Droplet Sizes Distribution (DSD) and Backscattering (BS) profiles evolution over 30 days were conducted, and their results are shown in Figure 5.

These two methodologies complement each other; BS can distinguish between oil droplet flocculation/coalescence and particle migration processes, whereas DSD analysis can provide information about changes in droplet size and/or flocculation’s presence. DSD was obtained in the presence and absence of Sodium Dodecyl Sulphate (SDS) solution to investigate the occurrence of flocculation during emulsification. The DSD remained constant either in the presence or absence of SDS which highlighted the absence of flocculation in the systems studied, apart from MG mayonnaise. In all cases, unimodal distributions were obtained, with the peak of smaller droplets located around 13 μm to nectarine mayonnaise, and 19 μm to standard mayonnaise. MG is the only sample that shows a bimodal distribution with a peak of smaller droplets at around 4 μm and a second broad peak at 130 μm, corresponding to another population of larger drops [25]. Bimodal emulsions show poorer stability than unimodal emulsions, meaning MG has lower stability [45]. An aspect to highlight about this second drop in population is that a tendency to decrease with time is observed, especially recently prepared and after 30 days of aging. This decrease could be interpreted as a consequence of a coalescence phenomenon that leads to the formation of larger droplets with diameters outside the measurement range of the equipment. In fact, with time, in this emulsion, you can see the formation of a thin layer of oil on its surface.

By looking at the standards and nectarine’s backscattering profiles, it is possible to understand that these mayonnaises are stable products since all profiles along the 30 days are overlaid on one curve [46]. On the other hand, MG shows destabilization phenomena of creaming. This instability is shown by a decrease in the BS values at the bottom (see Figure 5b, the red circles), while high values are obtained in the upper section due to the increased presence of oil droplets capable of reflecting the emitted light [10,40].

Figure 6 shows the results from the droplet size distribution measurements from day 1 and day 30, using a laser diffraction method for each curve, and the appearance of the respective droplets by microscopy using an oil immersion technique.

When DSD profiles, using SDS, for mayonnaises recorded on the first day are compared to those evaluated 30 days later, it is possible to infer that these samples are quite stable. The only exception is MG which shows different behaviors at the beginning and end of the day. This can be associated with the instability processes mentioned above.

Emulsifier type and concentration, particle size, oil concentration, and aqueous phase viscosity have all been shown to influence microstructural properties [35]. All samples showed a fine and well-dispersed oil-in-water structure in which fat globules were spherical, despite presenting droplets of different sizes (polydispersity).

Figure 7 and Table 8 show the Sauter (*d*_3,2_) and volumetric (*d*_4,3_) diameters, respectively, as a function of storage time of up to 30 days since production. The Sauter diameter is used as an average diameter for most droplets, whereas the volumetric diameter is associated with droplet size modifications related to destabilization processes, such as coalescence [25]. The results showed a reduction in the *d*_3,2_ and *d*_4,3_ values for NCTM with respect to the standard and MG. MG presents a diameter in the range of 5 μm, which may be related to the emulsifier used to produce this mayonnaise, which is egg yolk [47].

In general, all mayonnaises presented diameters ranging from 1 to 11 μm, values that are in the range of 100 nm to 100 μm for a conventional emulsion [48]. NCTM presented a Sauter diameter close to 1 μm, indicating certain stability, considering that small droplet sizes are related to stable emulsions [25]. Nevertheless, this sample does not show any significant differences from Mendes Gonçalves mayonnaise (*p* > 0.05).

NCTM shows great stability against coalescence, as *d*_4,3_ remains constant along storage time and since it does not show any significant differences across the thirty days. In standard mayonnaise, it is possible to see a slight decrease in *d*_4,3_ from day 1 to day 4 (*p* < 0.05), even so, the BS profile shows a stable emulsion, meaning that these instabilities were not significant for this mayonnaise [49]. MG shows differences across the period under study, meaning it suffered a destabilization phenomenon. Coalescence is noted in Table 8 for MG since from day 1 to day 4 the *d*_4,3_ increases significantly (p < 0.05). As previously mentioned, the decrease observed in this system for days 15 and 30 could be associated with the formation of larger droplets with diameters outside the measurement range of the equipment.

#### 3.1.6. Microbiological Analysis

According to Saraiva et al. (2019) [50], there are four groups of ready-to-eat foods, with mayonnaise belonging to Group 1B defined as “Fully cooked food, handled after heat treatment”. The interpretation of results obtained in microbiological tests on ready-to-eat foods depends on the number of colony-forming units (UFC) per gram or milliliter in the sample analyzed and classifies them into three levels: (i) satisfactory when the analytical result is within values; (ii) questionable when the analytical result is higher than the mean reference value and less than or equal to the maximum permissible value; (iii) not satisfactory when the analytical result is greater than the maximum permissible value. In Table 9, it is possible to observe the quantification of the different microorganisms that standard and nectarine mayonnaise were subjected to in the microbiological analysis.

After analyzing the obtained results, it is possible to conclude that nectarine flour contributes to reducing the growth of mesophiles compared to the control sample. Nevertheless, according to the guide values presented by Instituto Nacional de Saúde Doutor Ricardo Jorge (INSA), the Portuguese official laboratory for health affairs, both samples present satisfactory results since their values are lower than 10^4^ UFC/mL. In order to obtain satisfactory results for the quantification of yeasts and molds, the samples must present values lower than 10^3^ UFC/mL, which is in accordance with the findings of Table 9 [50]. Both *E. coli* and lactic acid bacteria results were less than 10 CFU/mL in the two samples, indicating a satisfactory outcome, meaning that their presence has not been detected. *Staphylococcus aureus* is a pathogenic bacterium that poses a potential health risk to consumers, as many strains produce enterotoxins, causing food poisoning when ingested [51]. According to INSA, for a food product to have a satisfactory result, its value should be less than 10 CFU/mL. Thus, it is possible to note that both samples presented unsatisfactory results. Even in the presence of potassium sorbate, standard mayonnaise could not control the spread of this bacterium, the values of which could be explained by the absence of good hygiene practices when producing the mayonnaises.

The *Staphylococcus aureus* values were not satisfactory for standard mayonnaise and NCTM was more notorious for the presence of this microorganism in NCTM. However, overall, nectarine flour allowed a reduction in the spread of different types of microorganisms compared to standard mayonnaise that had a traditional preservative (potassium sorbate) in its formulation, which represents a promising result. Bioactive compounds that are present in fruits also have a significant role as antimicrobial agents. In vitro procedures are used to characterize the ability to inhibit a certain microorganism [52]. Most studies are focused on inhibiting pathogenic microorganisms. Martínez et al. (2019) [53] studied the antimicrobial activity of olive extract rich in hydroxytyrosol (dilutions using 30–90 µL of extract). In the end, it was possible to observe increasing values of the inhibition radius of an olive extract rich in hydroxytyrosol when exposed to *Listeria monocytogenes* and *Staphylococcus aureus*. On the other hand, when in contact with *Escherichia coli*, it showed a limited effect of inhibition when using a low dilution (90 µL of extract).

## 4. Conclusions

The present work aimed to develop a mayonnaise with nutritional and sensory characteristics like the standard sample but without the use of additives. Fruit flours, especially nectarine flour, showed high potential as food ingredients, not only due to their antioxidant capacity but also thanks to their technological functionalities. Therefore, it is possible to introduce this ingredient into various types of food products, leading to a reduction in food waste.

By incorporating different fruit flours, it was possible to study the impact that this clean label ingredient had on the final product, in terms of pH, rheological properties, texture, antioxidant potential, microbiology, and stability.

All samples with the addition of fruit flour showed a pH lower than the standard sample, indicating a greater preservation capacity. By using fruit flour in mayonnaise, it was possible to produce mayonnaises with a higher degree of structuring compared to the standard. In addition, the values of firmness and adhesiveness of fruit mayonnaises, especially nectarine mayonnaise, were better than the standard sample. Through nutritional analyses, it was determined that all fruit flours have a higher antioxidant potential compared to the additives that are present in the standard sample. However, when fruit flour is added to the mayonnaise, it loses its antioxidant potential due to the complex matrix that mayonnaise presents; nevertheless, its antioxidant potential always remains superior in comparison to the standard sample. Microbiological analyses concluded that nectarine mayonnaise has a promising preservative capacity, acting as a clean label ingredient. However, standard mayonnaise and nectarine mayonnaise showed unsatisfactory results in the quantification of *Staphylococcus aureus*, indicating that they cannot control Gram-positive bacteria. All samples presented a unimodal droplet size distribution, except MG, which showed a bimodal distribution, leading to a more physically unstable product.

In the future, it would be interesting to produce nectarine mayonnaise under pilot-scale conditions, and also carry out long-term microbiology studies to understand if it will be possible to replace preservatives with clean label ingredients, such as fruit flour.

## Figures and Tables

**Figure 1 foods-12-02111-f001:**
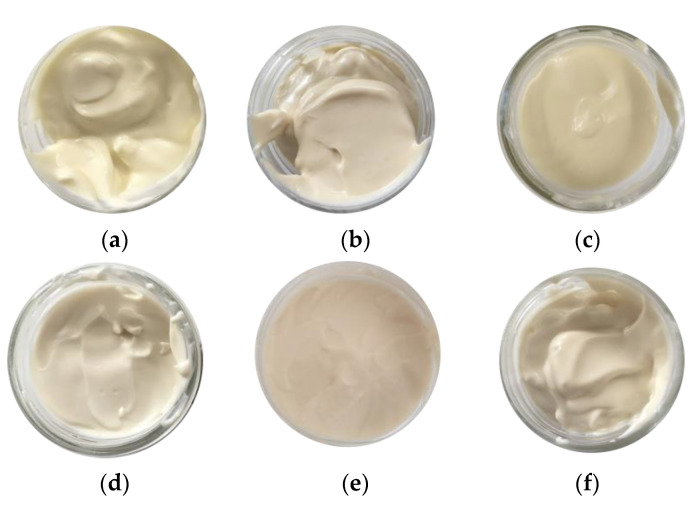
(**a**) Standard mayonnaise; (**b**) Mendes Gonçalves mayonnaise (MG); (**c**) Apple mayonnaise (APPM); (**d**) Nectarine mayonnaise (NCTM); (**e**) Pear mayonnaise (PRM); (**f**) Peach mayonnaise (PCHM).

**Figure 2 foods-12-02111-f002:**
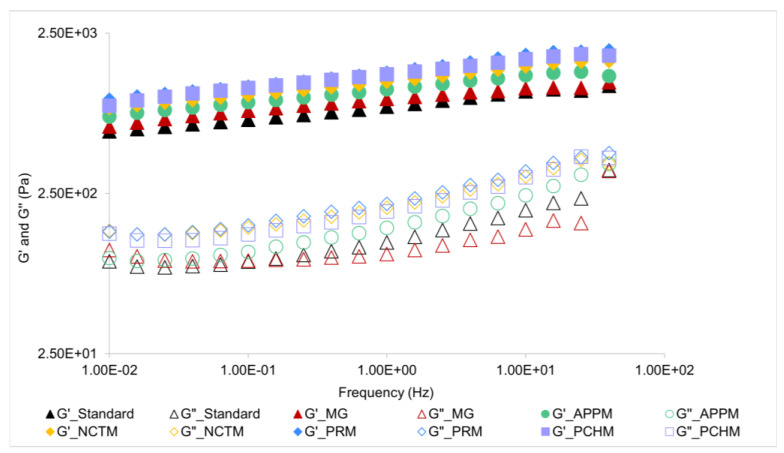
Mechanical spectrum of standard mayonnaise, Mendes Gonçalves mayonnaise (MG), and fruit mayonnaises—Apple mayonnaise (APPM), Nectarine mayonnaise (NCTM), Pear mayonnaise (PRM), and Peach mayonnaise (PCHM).

**Figure 3 foods-12-02111-f003:**
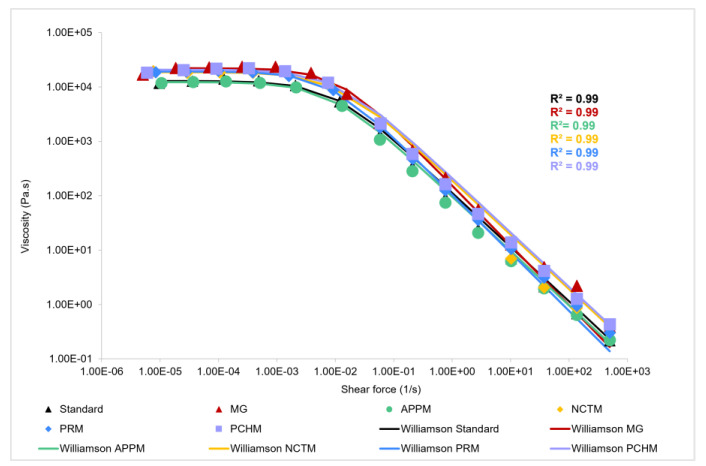
Flow curves of standard mayonnaise, Mendes Gonçalves mayonnaise (MG), and fruit mayonnaises—Apple mayonnaise (APPM), Nectarine mayonnaise (NCTM), Pear mayonnaise (PRM), and Peach mayonnaise (PCHM).

**Figure 4 foods-12-02111-f004:**
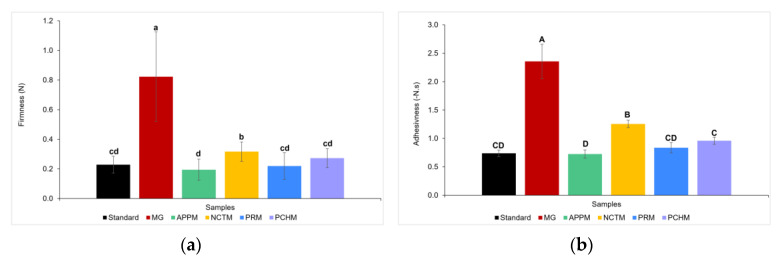
(**a**) Firmness, and (**b**) Adhesiveness of standard mayonnaise, Mendes Gonçalves mayonnaise (MG), and fruit mayonnaises—Apple mayonnaise (APPM), Nectarine mayonnaise (NCTM), Pear mayonnaise (PRM), and Peach mayonnaise (PCHM). Samples marked with different letters showed significant differences (*p* < 0.05, one-way ANOVA) for each parameter.

**Figure 5 foods-12-02111-f005:**
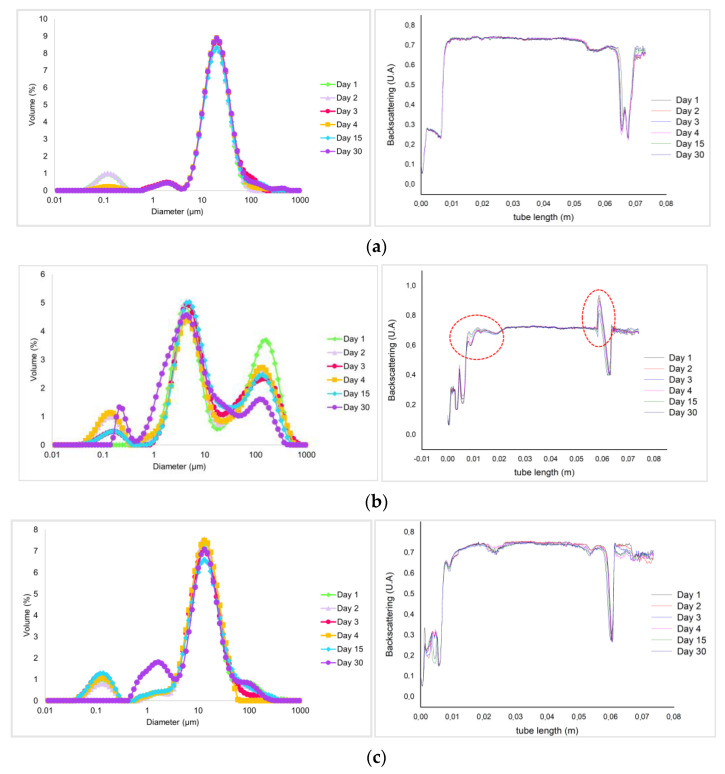
Droplet size distribution (**left**) and Backscattering (BS) profile (**right**) of (**a**) Standard mayonnaise; (**b**) Mendes Gonçalves mayonnaise (MG); and (**c**) Nectarine mayonnaise (NCTM).

**Figure 6 foods-12-02111-f006:**
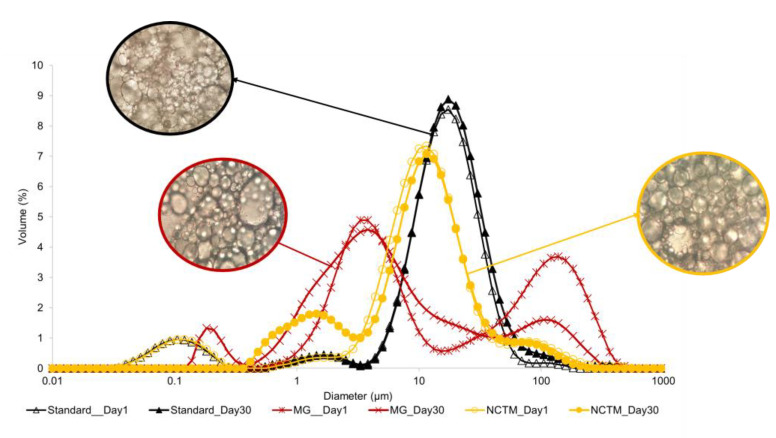
Droplet size distribution of Standard, Mendes Gonçalves (MG), and Nectarine (NCTM) mayonnaises, measured 1 and 30 days after preparation, but also oil droplet microscopy of mayonnaise samples with a magnification of 100×.

**Figure 7 foods-12-02111-f007:**
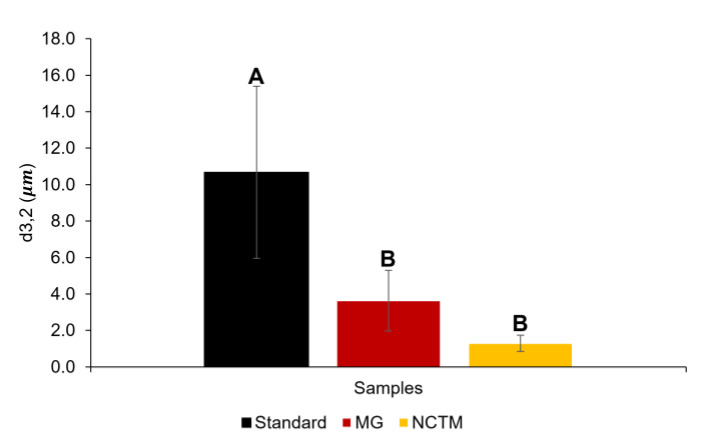
Average Sauter diameter (*d*_3,2_) along the thirty days for standard, Mendes Gonçalves (MG), and Nectarine (NCTM) mayonnaises. Samples marked with different letters showed significant differences (*p* < 0.05, one-way ANOVA) for each parameter.

**Table 1 foods-12-02111-t001:** Nutritional composition of apple, pear, nectarine, and peach flours obtained from the technical data sheets of the supplier, AgroGrIN Tech.

Flour	pH	Minerals (g)	Protein (g)	Fat (g)	Carbohydrates (g)	Simple Sugars (g)	Fiber (g)
Apple	4.00	1.88	2.60	0.96	80.78	54.50	10.35
Nectarine	4.35	3.51	5.60	0.60	79.83	44.30	7.22
Pear	3.50–4.00	1.66	2.50	1.00	72.66	26.90	17.80
Peach	5.47	2.71	4.70	0.40	80.21	38.30	8.31

**Table 2 foods-12-02111-t002:** Composition of Mendes Gonçalves commercial mayonnaise, standard mayonnaise, and mayonnaise with fruit flour.

Ingredients	Mendes Gonçalves Commercial Mayonnaise	Standard Mayonnaise	Fruit Mayonnaise
Water	√	√	√
Sunflower oil	√	√	√
Refined salt	√	√	√
Rosemary Extract	√	√	√
Concentrated Lemon Juice	√	√	√
Alcohol Vinegar	√	√	√
Egg yolk	√	x	x
Modified starch	√	x	x
Fine granulated sugar	√	√	x
Potassium sorbate	√	√	x
β-carotene	√	√	x
Vegetable protein	x	√	√
Fruit flour	x	x	√

**Table 3 foods-12-02111-t003:** Microbiological analysis made on mayonnaise samples.

Microbiological Analysis	Method	Reference
Coagulase-positive *Staphylococcus* count	Colony-count technique using Baird-Parker agar medium	ISO 6888-1:1999
Count of mesophiles at 30 °C	Colony count at 30 °C by the pour plate technique	ISO 4833-1:2013
Count of yeasts and molds	Colony count technique in products with water activity greater than 0.95	ISO 21527:2008
Count of lactic bacteria	Colony-count technique at 30 °C	ISO 15214:1998
Count of *Escherichia coli*	Detection and most probable number technique using 5-bromo-4-chloro-3-indolyl-ß-D-glucuronide	ISO 16649-3:2015

**Table 4 foods-12-02111-t004:** pH measurements, and the CIELAB colour parameters of standard mayonnaise, Mendes Gonçalves mayonnaise (MG), apple mayonnaise (APPM), nectarine mayonnaise (NCTM), pear mayonnaise (PRM), and peach mayonnaise (PCHM).

Mayonnaise Samples	pH	*L**	*a**	*b**	Δ*E**	Δ*E**
Standard	4.07 ± 0.00 ^c^	83.38 ± 0.43 ^b^	−2.37 ± 0.06 ^E^	12.12 ± 0.49 ^x^	-	8.28
MG	3.70 ± 0.01 ^a^	85.80 ± 0.64 ^a^	0.59 ± 0.24 ^A^	19.45 ± 1.77 ^w^	8.33	-
APPM	3.69 ± 0.02 ^a^	81.58 ± 0.30 ^c^	−1.11 ± 0.05 ^D^	11.51 ± 0.19 ^x^	2.29	9.15
NCTM	3.78 ± 0.00 ^b^	83.28 ± 0.14 ^b^	−0.81 ± 0.07 ^C^	9.43 ± 0.09 ^y^	3.12	10.43
PRM	3.79 ± 0.03 ^b^	80.18 ± 0.14 ^d^	−0.26 ± 0.07 ^B^	11.21 ± 0.06 ^x^	3.94	10.01
PCHM	3.71 ± 0.01 ^a^	81.54 ± 0.26 ^c^	−0.62 ± 0.02 ^C^	11.29 ± 0.30 ^x^	2.68	9.28

Samples marked with different letters showed significant differences (*p* < 0.05, one-way ANOVA) for each parameter.

**Table 5 foods-12-02111-t005:** Rheological parameters of standard mayonnaise, Mendes Gonçalves mayonnaise (MG), and fruit mayonnaises—Apple mayonnaise (APPM), Nectarine mayonnaise (NCTM), Pear mayonnaise (PRM), and Peach mayonnaise (PCHM), obtained from the mechanical spectra (G′ at 1 Hz and G^0^_N_), and from viscosity curves (*η*_0_).

Sample	G′ at 1 Hz (Pa)	Plateau Modulus G^0^_N_ (Pa)	*η*_0_ (× 10^4^ Pa.s)
Standard	868.07 ± 42.65 ^c^	701.43 ± 29.26 ^C^	1.05 ± 0.29 ^d^
MG	935.90 ± 60.28 ^c^	907.23 ± 57.62 ^ABC^	2.30 ± 0.10 ^a^
APPM	1090.40 ± 101.17 ^bc^	846.07 ± 74.14 ^BC^	1.30 ± 0.07 ^cd^
NCTM	1276.00 ± 30.64 ^ab^	992.33 ± 30.64 ^AB^	1.71 ± 0.14 ^bc^
PRM	1365.67 ± 128.91 ^a^	1064.83 ± 113.35 ^A^	1.88 ± 0.04 ^ab^
PCHM	1378.33 ± 152.19 ^a^	1091.57 ± 113.19 ^A^	1.69 ± 0.34 ^bc^

Samples marked with different letters showed significant differences (*p* < 0.05, one-way ANOVA) for each parameter.

**Table 6 foods-12-02111-t006:** Antioxidant capacity by FRAP, DPPH, and Total Phenolic Compounds (mg equivalent of gallic acid/100 g) of fruit flours, as well as the standard sample (STF) composed of sugar, preservative, and colorant.

Sample	DPPH	FRAP	TPC
Sugar, potassium sorbate, colorant (STF)	0.86 ± 0.15 ^d^	3.62 ± 0.23 ^D^	49.89 ± 23.05 ^y^
Apple Flour (APPF)	39.45 ± 1.63 ^b^	60.62 ± 3.08 ^B^	170.49 ± 7.80 ^x^
Nectarine Flour (NCTF)	28.31 ± 1.39 ^c^	53.37 ± 6.44 ^BC^	157.27 ± 14.21 ^x^
Pear Flour (PRF)	88.41 ± 1.75 ^a^	106.13 ± 5.04 ^A^	167.81 ± 71.83 ^x^
Peach Flour (PCHF)	28.88 ± 1.80 ^c^	46.08 ± 1.25 ^C^	172.83 ± 25.45 ^x^

Samples marked with different letters showed significant differences (*p* < 0.05, one-way ANOVA) for each parameter.

**Table 7 foods-12-02111-t007:** Antioxidant capacity by FRAP, and DPPH, and Total Phenolic Compounds (mg equivalent of gallic acid/100 g) of standard mayonnaise and fruit mayonnaises—Apple mayonnaise (APPM), Nectarine mayonnaise (NCTM), Pear mayonnaise (PRM), and Peach mayonnaise (PCHM).

Sample	DPPH	FRAP	TPC
Standard	2.38 ± 0.39 ^b^	5.55 ± 0.09 ^B^	28.96 ± 7.90 ^x^
APPM	4.21 ± 0.56 ^a^	11.87 ± 0.57 ^A^	20.01 ± 2.43 ^x^
NCTM	2.87 ± 0.21 ^b^	11.30 ± 0.19 ^A^	25.56 ± 8.35 ^x^
PRM	4.48 ± 0.35 ^a^	12.06 ± 0.21 ^A^	22.94 ± 0.73 ^x^
PCHM	2.94 ± 0.26 ^b^	11.75 ± 0.25 ^A^	29.81 ± 4.61 ^x^

Samples marked with different letters showed significant differences (*p* < 0.05, one-way ANOVA) for each parameter.

**Table 8 foods-12-02111-t008:** Volumetric diameter (*d*_4,3_) of standard, Mendes Gonçalves (MG), and Nectarine (NCTM) mayonnaises.

Sample	*d*_4,3_ (μm)			
	Day 1	Day 4	Day 15	Day 30
Standard	24.56 ± 1.11 ^A^	21.39 ± 0.34 ^B^	25.09 ± 0.89 ^A^	26.27 ± 0.23 ^A^
Mendes Gonçalves mayonnaise	45.29 ± 2.91 ^b^	56.23 ± 3.50 ^a^	44.01 ± 0.28 ^b^	27.58 ± 1.31 ^c^
Nectarine Mayonnaise	20.15 ± 0.99 ^M^	19.01 ± 1.50 ^M^	21.09 ± 1.22 ^M^	19.33 ± 1.52 ^M^

Samples marked with different letters showed significant differences (*p* < 0.05, one-way ANOVA) for each parameter.

**Table 9 foods-12-02111-t009:** Quantification of the different microorganisms from the microbiological analysis made on Standard and Nectarine mayonnaises (NCTM).

Sample	Mesophiles	Molds and Yeasts	*E. coli*	*Staphylococcus aureus* Coagulase +	Lactic-Acid Bacteria
Standard Mayonnaise	8.3 × 10^3^ UFC/mL	8.0 × 10^2^ UFC/mL	<10 UFC/mL	<100 UFC/mL	<10 UFC/mL
Nectarine Mayonnaise	2.9 × 10^2^ UFC/mL	1.5 × 10^2^ UFC/mL	<10 UFC/mL	1.5 × 10^2^ UFC/mL	<10 UFC/mL

## Data Availability

The data presented in this study are available on request from the corresponding author.

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
