# Peer review of "Development of a Clean Label Mayonnaise Using Fruit Flour"

_foods, 2023, doi:10.3390/foods12112111_

Round 1

Reviewer 1 Report

The article presents the properties of clean label mayonnaises based on fruit flours. However, it requires some corrections.

Below are some tips/comments about the paper:

L18-20.”… mayonnaises produced with fruit flour had better structure parameters…” Please write down exactly what parameters you mean.

L41. Please change "o/w" to "oil-in-water (o/w)".

L46-46. Emulsifying properties have egg yolk not a whole egg. Egg yolk is not animal protein. I suggest replacing “animal protein” to “animal source” and “egg” to “egg yolk”.

L68-73. Please cite literature for these statements.

L79-81. Preservatives? Traditional mayonnaise with a fat content of 65-80% does not contain preservatives. It is a safe product due to its high fat content and the presence of acidic ingredients. This is legally guaranteed in most countries.

L101-102. This sentence should be rewritten. Please provide the composition of standard mayonnaise, commercial mayonnaise and mayonnaise with fruit flour. And them compare it.

L105-108. The composition of mayonnaises should be presented. The lack of mayonnaise composition makes it difficult to read the text. Providing a reference to the literature [16] is not enough. The authors did not present the contribution of fruit flour in the mayonnaise recipe. Why did the authors decide to add potassium sorbate? Which samples? High fat content and the presence of acidic ingredients affect the stability of mayonnaise. In the authors' earlier publication [16], there is no β-carotene in the recipe of mayonnaise. Was the composition of the commercial mayonnaise identical to the standard mayonnaise, with the exception of replacing the yolk with vegetable protein?

L137-154. Please specify the temperature at which the rheological measurements were carried out.

L142-143. Expand symbol abbreviations G’ and G’’.

L242-243. Add unit for GAE.

L249 and elsewhere. There is a problem in the text when referring to figures or tables. Needs improvement.

L260-261. Was the replacement of sugar, potassium sorbate and β-carotene with flour in the mayonnaise recipe identical? What was the reason for using potassium sorbate? And why only in a standard mayonnaise recipe? How can fruit flour replace a preservative? Give literature on this fact.

Eq. 2 and L156. There should be "n" in the equation instead of "m", the same in the text. And "n<1" for shear thinning fluids.

L273-275. The word "a highly" is exaggerated in the sentence. Please remove.

L276-278. Please replace “are now” with a word “were”.

L290-299. To add more discussion, please compare the color results with those of other scientists.

Figure 2-6. Add the axis marks of the main unit.

Figure 2. Please change the values on the OX axis and the OY axis without the "E".

L314-315. Unclear. Please rewritten this sentence.

L318-321. Style. Please reformulation this sentence.

Figure 3. Shear force? It should be “Shear rate”. Please change the values on the OX axis and the OY axis without the "E".

Figure 4a. How will the authors explain such a large standard deviation of the results for Mendes Gonçalves (MG) mayonnaise? Its recipe was based on commercial mayonnaise available on the market?

L369-370. Style. Please reformulation this sentence.

L406-407. Control sample? Which one?

Table 6. Why are the results for Mendes Gonçalves (MG) mayonnaise not shown in table 6?

L476-477. Style. Please reformulation this sentence.

L574-576. Unclear. Please reformulation this sentence.

L576-579. Unclear. Please reformulation this sentence.

L585-587. Both samples? Which samples are you talking about exactly?

Article is written in understandable language.

Sometimes there are incorrectly stylistic sentences that make it difficult to interpreted the text or wrong tense is used in sentences.

"Although", "However" it should be used at the beginning of the sentence. 

Author Response

Reviewer #1

Comments and Suggestions for Authors

Dear reviewer, we appreciate your kind words. We took your advice in consideration. All alterations in the text are highlighted in yellow for your convenience.

The article presents the properties of clean label mayonnaises based on fruit flours. However, it requires some corrections. 

Below are some tips/comments about the paper:

  1. L18-20.” mayonnaises produced with fruit flour had better structure parameters…” Please write down exactly what parameters you mean.

This aspect was clarified in the text: (…) better structure parameters in terms of viscosity, frequency and texture, but also improved pH and antioxidant activity

  1. Please change "o/w" to "oil-in-water (o/w)".

Reviewed

  1. L46-46. Emulsifying properties have egg yolk not a whole egg. Egg yolk is not animal protein. I suggest replacing “animal protein” to “animal source” and “egg” to “egg yolk”.

Reviewed

  1. L68-73. Please cite literature for these statements.

Reviewed: This ingredient provides health benefits due to its high fibre and antioxidant content. Through its use, it is possible to incorporate bioactive compounds, like carotenoids and phenolic acids in mayonnaises, increasing stability and shelf-life of the final products, while enhancing the product’s colour and increasing potential health benefits (Santos et al., 2022). Potential texture enhancing capacity is expected due to its dietary fibre content, which impacts the water holding and retention capacity, oil holding capacity, foam capacity and stability (Santos et al., 2022).

  1. L79-81. Preservatives? Traditional mayonnaise with a fat content of 65-80% does not contain preservatives. It is a safe product due to its high fat content and the presence of acidic ingredients. This is legally guaranteed in most countries. 

Preservatives are presented in the commercial mayonnaise of the company. Since the main objective of the present work was to produce a clean label mayonnaise based on the commercial mayonnaise, but also with the standard one, we took this ingredient and substitute it with fruit flour.

  1. L101-102. This sentence should be rewritten. Please provide the composition of standard mayonnaise, commercial mayonnaise and mayonnaise with fruit flour. And them compare it.

The sentence was rewritten: “All fruit mayonnaises were later compared to standard mayonnaise (mayonnaise without fruit flour) and with MG produced on a laboratory scale, to have the same emulsifying scale.”

  1. L105-108. The composition of mayonnaises should be presented. The lack of mayonnaise composition makes it difficult to read the text. Providing a reference to the literature [16] is not enough.
    1. The authors did not present the contribution of fruit flour in the mayonnaise recipe. Why did the authors decide to add potassium sorbate? Which samples? High fat content and the presence of acidic ingredients affect the stability of mayonnaise.
    2. In the authors' earlier publication [16], there is no β-carotene in the recipe of mayonnaise.
    3. Was the composition of the commercial mayonnaise identical to the standard mayonnaise, with the exception of replacing the yolk with vegetable protein?
    4. The composition of the mayonnaise with the amounts of ingredients cannot be presented due to confidentiality issues. The composition of commercial mayonnaise is the one found in the market. However, a table was introduced to clarify the composition of each mayonnaise.

7.a) Fruit flour was added to the mayonnaise on the same percentage that the some of potassium sorbate, sugar and colorant had on the standard mayonnaise. Potassium sorbate was present on the recipe since the commercial mayonnaise of the company presents this ingredient. Thus, potassium sorbate was present on the standard sample and Mendes Gonçalves commercial mayonnaise.

7.b) Regarding authors' earlier publication [16], there is β-carotene in the recipe “(…) foodgrade

refined NaCl >99%(w/w) purity; food-grade sucrose >99,5%(w/w) purity; lemon juice (concentrated by a factor of 7); 8–10%(v/v) acetic acid alcohol vinegar; and food-grade β-carotene E160a (i).

7.c) The difference between those two mayonnaises, is that standard mayonnaise does not present egg yolk neither modified starch.

  1. L137-154. Please specify the temperature at which the rheological measurements were carried out.

Reviewed: 20 ± 0.5°C

  1. L142-143. Expand symbol abbreviations G’ and G’’.

Reviewed: “(…) which means, storage modulus (G’), and loss modulus (G’’) as a function of frequency. “

  1. L242-243. Add unit for GAE.

Reviewed.

  1. L249 and elsewhere. There is a problem in the text when referring to figures or tables. Needs improvement.

It was corrected.

  1. L260-261. Was the replacement of sugar, potassium sorbate and β-carotene with flour in the mayonnaise recipe identical? What was the reason for using potassium sorbate? And why only in a standard mayonnaise recipe? How can fruit flour replace a preservative? Give literature on this fact.

The replacement of sugar, potassium sorbate and β-carotene with fruit flour was identical, and the reason for using preservative, is because it belongs to the original recipe.

Regarding the literature for the replacement of fruit flour with a preservative, there is a scientific paper, from (Munekata et al., 2023) that summarizes some studies regarding this subject. I added some information that is present on this paper on the present work. 

  1. 2 and L156. There should be "n" in the equation instead of "m", the same in the text. And "n<1" for shear thinning fluids.

It was corrected.

  1. L273-275. The word "a highly" is exaggerated in the sentence. Please remove.

It was corrected.

  1. L276-278. Please replace “are now” with a word “were”.

It was corrected.

  1. L290-299. To add more discussion, please compare the color results with those of other scientists.

The results were compared to other scientists.

  1. Figure 2-6. Add the axis marks of the main unit.

The axis marks were introduced.

  1. Figure 2. Please change the values on the OX axis and the OY axis without the "E".

On Excel it is not possible to do that. I can make it handwritten but it will lose quality.

  1. L314-315. Unclear. Please rewritten this sentence.

It was rephrased and the sentence was completed.

  1. L318-321. Style. Please reformulation this sentence.

It was rephrased and the sentence was completed.

  1. Figure 3. Shear force? It should be “Shear rate”. Please change the values on the OX axis and the OY axis without the "E".

The figure was changed. On Excel it is not possible to do that. I can make it handwritten but it will lose quality.

  1. Figure 4a. How will the authors explain such a large standard deviation of the results for Mendes Gonçalves (MG) mayonnaise? Its recipe was based on commercial mayonnaise available on the market?

There is a big difference with MG mayonnaise and the remaining samples, since MG has egg yolk on its composition. Its recipe is the same as the mayonnaise available on the market, but the one produced for the present work was done on a laboratory scale.

  1. L369-370. Style. Please reformulation this sentence.

It was rephrased and the sentence was completed.

  1. L406-407. Control sample? Which one?

Standard mayonnaise. The sentence was changed to be more clear.

  1. Table 6. Why are the results for Mendes Gonçalves (MG) mayonnaise not shown in table 6?

On this particular study, MG was not studied since we wanted to quantify the amount of antioxidants that the flours could give to the mayonnaise. Since standard mayonnaise, had the same formulation as the fruit mayonnaises, MG was not studied.

  1. L476-477. Style. Please reformulation this sentence.

It was rephrased and the sentence was completed.

  1. L574-576. Unclear. Please reformulation this sentence.

It was rephrased and the sentence was completed.

  1. L576-579. Unclear. Please reformulation this sentence.

It was rephrased and the sentence was completed.

  1. L585-587. Both samples? Which samples are you talking about exactly?

Standard mayonnaise and Nectarine mayonnaise. Sentence was rephrased to be clearer.

Dear reviewer, once again, we appreciate all your feedback and took it in consideration to improve our manuscript. Best regards.

Reviewer 2 Report

Dear authors, 

After reading the manuscript "Development of a clean label mayonnaise using fruit flour", I realized that the manuscript showed in some parts the scientific rigour wanted, but in other parts I have missed it.

The authors have presented critical evaluation only in some paragraphs.

The references are not exactly current, besides the objective and conclusion can be improved.

Thats why I have written some suggestions below in an attempt to improve the paper.

L.32- Standardize on journal rules =>  Park & Kim (2021) [1]

L. 40- I request attention to this quote because the authors mention "different definitions" "the Institute of Food Technologists" and quote only [2]. It gets unclear to understand. Please, improve it here.

L.41- I missed the information about the consumption of the chosen product (mayonnaise) in the world, and whether egg-free products are already easily found in different countries of the world.

L.47-48- You need to quote this paragraph => "Hen’s egg is a common food allergen but is also a 47 source of possible contamination with Salmonella sp"

L.49-Where is author [5] ?

L.57- It seems to me that you will have to review all the citation numbering. In here, FAO should be numbered and I checked and it is not any of those authors cited in the paragraph.

L.61- [10] => I also did not find it

L 79, 89, 97 and 102- It got too repetitive about Casa Mendes Gonçalves

L.90-94- A table with the amounts of ingredients would be very important.  "refined sunflower oil, potassium sorbate >98%(w/w) purity, fine granulated sugar, refined salt, lemon juice (concentrated by a factor of 7), rosemary  extract, 8–10%(v/v) acetic acid alcohol vinegar, β-carotene colourant, and vegetable proteins (lupin protein isolate with 87-95% protein content, and faba bean protein concentrate with 60% protein content)."

L.99- In my opinion this is not the best place to insert this table. For example, the methodology for pH will only be presented in L. 123.

L.106 - See authors' guidelines about authors citation.

L.106- Where ? for place. in which ??

L.108- Why don't you use initials to differentiate the mayonnaises ( instead of "both mayonnaises") and not keep repeating Casa Mendes Gonçalves.

L.133- where ? For place. Please, check all of them in the paper.

L.264- (Error! Reference source not found.) 

L.274- (Error! Reference source not found.)

L.269- Please, use this initials in the whole paper.

L.269- Authors, follow  the same order for flours inthe text and tables, figure. In table 1 the order is different from figure 1, for instance.

L.282- Authors' guideline for citation

L.287- I'm sorry,but it was not clear to me why 2 columns for Comparing the total colour difference (E*)

L.306- Error! Reference source not found.

L.344- Error! Reference source not found.

L.353- Error! Reference source not found.

L.369- Error! Reference source not found.

L.371- Why  is sometimes standard before MG? standardise, please.

Why capital letter in figure 4b ?

L.390- Error! Reference source not found.

L.396- why STF ? What about standard and MG ? Does it need  2 tables (5 and 6) ? I found it confusing 

L.407- Error! Reference source not found.

L.420- Error! Reference source not found.

L.432- Error! Reference source not found.

L.523- Error! Reference source not found.

L.571- In my opinion it would be interesting if all treatments were evaluated from beginning to end.

The clean label appears in the title and conclusion, but I think it needs to be re-evaluated with the study staff if the clean label is really focused on your study,. Sorry, but I see the same analyses performed in conventional studies on the chemical, technological and microbiological analyses, not exactly on clean label. I missed the sensory analysis, by the way.

L.576 and 581- Conclusion is to answer your objectives. The presentation of statistical difference should be limited to the results.

Moderate editing of English language

 English is always useful to ask a native speaker for a final appreciation before submitting.

Author Response

Reviewer #2

Comments and Suggestions for Authors

Dear reviewer, we appreciate your kind words. We took your advice in consideration. All alterations in the text are highlighted in yellow for your convenience.

  1. 32- Standardize on journal rules =>  Park & Kim (2021) [1]

A reference manager software was used to check all references, they should now be correct. In some cases, a name and year are referred for phrasing purposes, but the correct reference inside brackets is presented as well.

  1. 40- I request attention to this quote because the authors mention "different definitions" "the Institute of Food Technologists" and quote only [2]. It gets unclear to understand. Please, improve it here.

The sentence was rephrased: “However, there is no legal definition regarding this subject, leading to the creation of different definitions [2]. For example, the Institute of Food Technologists (…)”

  1. 41- I missed the information about the consumption of the chosen product (mayonnaise) in the world, and whether egg-free products are already easily found in different countries of the world.

This information was added in the introduction.

  1. 47-48- You need to quote this paragraph => "Hen’s egg is a common food allergen but is also a 47 source of possible contamination with Salmonella sp"

It was cited.

  1. 49-Where is author [5] ?

It was corrected.

  1. 57- It seems to me that you will have to review all the citation numbering. In here, FAO should be numbered and I checked and it is not any of those authors cited in the paragraph

The author was added to the text..

  1. 61- [10] => I also did not find it

It was corrected.

  1. L 79, 89, 97 and 102- It got too repetitive about Casa Mendes Gonçalves

The sentences were rephrased.

  1. 90-94- A table with the amounts of ingredients would be very important.  "refined sunflower oil, potassium sorbate >98%(w/w) purity, fine granulated sugar, refined salt, lemon juice (concentrated by a factor of 7), rosemary  extract, 8–10%(v/v) acetic acid alcohol vinegar, β-carotene colourant, and vegetable proteins (lupin protein isolate with 87-95% protein content, and faba bean protein concentrate with 60% protein content)."

The composition of the mayonnaise with the amounts of ingredients cannot be presented due to confidentiality issues. The composition of commercial mayonnaise is the one found in the market. However, a table was introduced to clarify, as much as possible, the composition of each mayonnaise.

  1. 99- In my opinion this is not the best place to insert this table. For example, the methodology for pH will only be presented in L. 123.

This table was not used for any study, it is just indicative, that’s why it is on the materials and methods.

  1. 106 - See authors' guidelines about authors citation.

A reference manager software was used to check all references, they should now be correct. In some cases, a name and year are referred for phrasing purposes, but the correct reference inside brackets is presented as well.

  1. 106- Where ? for place. in which ??

Reviewed.

  1. 108- Why don't you use initials to differentiate the mayonnaises (instead of "both mayonnaises") and not keep repeating Casa Mendes Gonçalves.

The initials were added to the text to avoid repeating the same words.

  1. 133- where ? For place. Please, check all of them in the paper.

Reviewed.

  1. 264- (Error! Reference source not found.) 

Reviewed

  1. 274- (Error! Reference source not found.)

Reviewed

  1. 269- Please, use this initials in the whole paper.

The initials were added to the text to avoid repeating the same words.

  1. 269- Authors, follow  the same order for flours inthe text and tables, figure. In table 1 the order is different from figure 1, for instance.

The order was changed to be more easily read.

  1. 282- Authors' guideline for citation

A reference manager software was used to check all references, they should now be correct. In some cases, a name and year are referred for phrasing purposes, but the correct reference inside brackets is presented as well.

  1. 287- I'm sorry,but it was not clear to me why 2 columns for Comparing the total colour difference (DE*)

MG and standard mayonnaises were used as the target and control samples. In this way, one column compares the standard mayonnaise with the remaining samples, whereas the other one compares MG with the remaining samples.

  1. 306- Error! Reference source not found.

Reviewed

  1. 344- Error! Reference source not found.

Reviewed

  1. 353- Error! Reference source not found.

Reviewed

  1. 369- Error! Reference source not found.

Reviewed

  1. 371- Why is sometimes standard before MG? standardise, please.

The order was changed to be more easily read.

  1. Why capital letter in figure 4b ?

It is in capital letter since it is a different parameter that is being studied.

  1. 390- Error! Reference source not found.

Reviewed

  1. 396- why STF ? What about standard and MG ? Does it need  2 tables (5 and 6) ? I found it confusing 

STF is composed by the ingredients that are substituted in the fruit mayonnaises, i.e., potassium sorbate, sugar, and salt. It has another name, since we are comparing the fruit flours and not the mayonnaises.

  1. 407- Error! Reference source not found.

Reviewed

  1. 420- Error! Reference source not found.

Reviewed

  1. 432- Error! Reference source not found.

Reviewed

  1. 523- Error! Reference source not found.

Reviewed

  1. 571- In my opinion it would be interesting if all treatments were evaluated from beginning to end.

The text was reviewed, and all treatments were considered on the conclusion of the present work.

  1. The clean label appears in the title and conclusion, but I think it needs to be re-evaluated with the study staff if the clean label is really focused on your study. Sorry, but I see the same analyses performed in conventional studies on the chemical, technological and microbiological analyses, not exactly on clean label. I missed the sensory analysis, by the way.

Your suggestion was considered and appreciated. This work is part of a Portuguese funded project: cLabel+ (POCI-01-0247-FEDER-046080, https://cleanlabelplus.pt/), a project that aims to redesign traditional food products, aligning them with the clean label trend. The whole work was designed to replace non-clean label food ingredients such as sugar and food colourant by clean label ingredients, fruit flours. What makes the work “clean label” is the replacement of traditional ingredients, often perceived by the consumer as artificial and prejudicial for health, by cleaner ingredients. Fruit flours, besides being natural sources of sugar, carotenoids (hence replacing the referred ingredients) and fibre, are also perfectly aligned with the sustainable foods trend, since they are obtained from out-of-format fruits, contributing to circular economy. Using naturally-perceived and sustainable ingredients is a common clean label approach. Hence, the authors consider that the work is aligned with that trend, thus considering correct the use of the term “clean label” along the manuscript, as well as on the title.

  1. 576 and 581- Conclusion is to answer your objectives. The presentation of statistical difference should be limited to the results.

The text was reviewed to remove statistical results from the conclusion.

Dear reviewer, once again, we appreciate all your feedback and took it in consideration to improve our manuscript. Best regards.

Round 2

Reviewer 2 Report

After another evaluation of the manuscript, I realized a great improvement in the quality of the paper.